# Alternatives in Education—Rat and Mouse Simulators Evaluated from Course Trainers’ and Supervisors’ Perspective

**DOI:** 10.3390/ani11071848

**Published:** 2021-06-22

**Authors:** Melanie Humpenöder, Giuliano M. Corte, Marcel Pfützner, Mechthild Wiegard, Roswitha Merle, Katharina Hohlbaum, Nancy A. Erickson, Johanna Plendl, Christa Thöne-Reineke

**Affiliations:** 1Institute of Animal Welfare, Animal Behavior and Laboratory Animal Science, Department of Veterinary Medicine, Freie Universität Berlin, 14163 Berlin, Germany; Mechthild.Wiegard@fu-berlin.de (M.W.); Katharina.Hohlbaum@fu-berlin.de (K.H.); NancyAnn.Erickson@fu-berlin.de (N.A.E.); Thoene-Reineke.Christa@fu-berlin.de (C.T.-R.); 2Institute of Veterinary Anatomy, Department of Veterinary Medicine, Freie Universität Berlin, 14195 Berlin, Germany; Giuliano.Corte@fu-berlin.de (G.M.C.); office@myhumanx.com (M.P.); Johanna.Plendl@fu-berlin.de (J.P.); 3Institute for Veterinary Epidemiology and Biostatistics, Department of Veterinary Medicine, Freie Universität Berlin, 14163 Berlin, Germany; Roswitha.Merle@fu-berlin.de; 4MF 3—Animal Facility—Method Development and Research Infrastructure, Robert Koch-Institute, 13353 Berlin, Germany

**Keywords:** 3R principle, humane education, training, alternative, laboratory animals, EU Directive, survey, SimulRATor

## Abstract

**Simple Summary:**

Simulators for training in laboratory animal science bear great potential to overcome the dilemma between the present demand for high-quality practical training involving live animals whilst implementing the “3R principle” (Replace, Reduce, Refine) according to the Directive 2010/63/EU. Currently, one mouse and six rat simulators are available, but only few data on them exist. To advance simulator-based training, an online survey for course trainers and supervisors of laboratory animal training courses focusing mice and rats was conducted, as these groups are most aware of its implementation due to applying alternative education and training methods regularly. This study reflects the current awareness, implementation, and satisfaction concerning methodical and practical criteria of the simulators including the requirements for a new development of 35 course trainers and supervisors who completed a German online survey conducted between May 2018 and June 2019. Although the study revealed a high awareness of existing simulators, their implementation is rather low, perhaps due to them not meeting certain demands. Generally, an approval of simulator-based training and a demand for user-optimized, realistic, financially affordable, and robust rat and mouse simulators were indicated, which may strongly benefit the 3Rs and animals in all experimental areas.

**Abstract:**

Simulators allow the inexperienced to practice their skills prior to exercise on live animals. Therefore, they bear great potential in overcoming the dilemma between the present demand for high-quality practical training involving live animals whilst implementing the 3R principle according to the Directive 2010/63/EU. Currently, one mouse and six rat simulators are commercially available. As data on their impact are lacking, this project aimed at providing an overview of the awareness, implementation, and methodical and practical satisfaction provided by 35 course trainers and supervisors of laboratory animal training courses for mice and rats regarding the simulators available. Although simulators facilitate training of relevant techniques and relatively high awareness of them seemed to be present, their implementation is currently very low, possibly due to lack of meeting the respondents’ demands. Thus, this study revealed the overall approval of simulator training and general demand for user-optimized, realistic, and financially affordable simulators and, hence, indicates a strong impulse for new developments strengthening the 3Rs as a benefit to all animals used in research.

## 1. Introduction

The 3R principle (Replace, Reduce, Refine) by Russell and Burch [1] has been implemented in the European Directive 2010/63/EU [2] which stipulates that alternatives to animal experiments must be implemented whenever possible. Thereby, existing models and methods are refined, study or animal numbers and discomfort or stress are reduced, and experiments are replaced whenever possible. For a sustainable implementation of the 3R principle, some authors consider education and training as perhaps one of the most effective approaches [3,4,5]. Furthermore, a high educational standard of all staff involved in animal experiments can minimize potential burden experienced by laboratory animals. This, in turn, may positively affect the robustness of data and can improve study outcomes and reproducibility. To ensure the protection of laboratory animals, the European Directive 2010/63/EU [2] requires essential knowledge and skills of respective persons prior to carrying out animal experiments which can be obtained in dedicated Laboratory Animal Science (LAS) courses.

Because animal-free methods alone are not sufficient to provide appropriate practical training, recommendations of specialist organizations, such as the Federation of European Laboratory Animal Science Associations, FELASA [6], or the Gesellschaft für Versuchstierkunde/Society of Laboratory Animal Science, GV-SOLAS [7], acknowledge practical training involving live animals for some techniques, in order to maintain high standards of education. The most commonly used animal species in experiments for scientific or pharmaceutical purposes and, hence, in LAS education and training, are rats and mice.

As pain, suffering, distress or lasting harm cannot be fully excluded in practical training, the European Directive 2010/63/EU classifies practical training on live animals as animal experiment and thus requires the implementation of the 3Rs [2].

Here, a dilemma of education arises between the demand for high-quality practical training involving live animals which benefits animals used in future experiments whilst the protection of these animals used in education is simultaneously required by law.

In order to minimize potential negative impact on the animals used for training, a systematic course design is favorable. The untrained person is first instructed theoretically using photos, videos, and demonstrations prior to non-animal practical training under competent supervision using toys, do-it-yourself-inventions, skin imitations for suturing or simulators of the whole animal’s body or parts of it. Based on our experience, we assume that for practicing LAS procedures particularly in small mammals, “whole-animal” simulators of the target species seem to allow for optimal skills practice and preparation due to their equivalent anatomy or size ratio. These characteristics are especially important for the practice of restraint which, in turn, is an essential prerequisite in minimizing distress during procedures and providing procedural success and reproducibility. Furthermore, they could potentially support all 3Rs concerning future technological developments.

The positive impact of simulator-based training has already been suggested for surgical techniques and dissection [8,9,10,11] as well as in medical [12,13,14] or veterinary education [15,16,17] and the educational efficacy of alternative teaching methods has been reviewed comprehensively by Zemanova and Knight [18]. However, data concerning the use of simulators in LAS education and training and its impact on the 3Rs are lacking. Only a few publications are available regarding a rat simulator specifically designed for microsurgical techniques [19,20,21,22], which was also assessed by the participants in this survey for course trainers and supervisors. Additionally, currently available simulators are mentioned in books, reviews, and reports concerning alternative learning methods [23,24,25,26].

For LAS courses focusing on mice and rats, one mouse and six rat simulators are, to our knowledge, currently available (Table 1). If training with simulators can maintain or even improve the high standard of education, these simulators seem to bear a great potential to expand the 3Rs to education and training.

Hence, to provide first scientific data with specific regard to simulator-based training in handling and procedural techniques on mice and rats, the aim of this study is to give an overview of LAS course trainers’ and supervisors’ actual awareness, implementation, and methodical and practical satisfaction with the available rat and mouse simulators. Furthermore, we aimed to determine methodical and practical requirements for a new simulator.

## 2. Materials and Methods

### 2.1. Design and Pretest of the Survey Protocol

The survey was designed and conducted via Limesurvey [28], targeting trainers and supervisors of mouse- and/or rat-based LAS courses of German-speaking countries.

Initially, factors concerning awareness, usage, satisfaction or requirements for simulators were determined. For each of these factors, a hypothesis and a respective question item was developed in regard to the available simulators and recommendations for LAS courses [6,7,29]. In accordance with questionnaire guidelines published by GESIS—Leibniz Institute for Social Sciences [30,31,32,33], the question type was firstly determined for each item with respect to the question’s aim. Secondly, the questions were further reworded according to the guidelines. Lastly, the order of the questions in the survey was arranged. In order to ensure the questionnaire’s comprehensibility and interpretability, it was assessed in three consecutive cognitive pretests [34] by LAS scientists, each followed by thorough revision.

The final survey protocol represented a mixed-mode questionnaire with 89 voluntary questions consisting of 67 closed-ended questions including multiple-choice, single-selection, 6-point Likert items, rankings—in which participants must order features according to their priority, and numerical input—and 22 open-ended questions (see Appendix A). The questions focused on four main topics: Firstly, information on the LAS courses and their organization; secondly, awareness, implementation, and assessment of the simulators used in their most frequently performed course type; thirdly, requirements for operational rat and mouse simulators; and finally, demographic information and feedback message. The participants only received questions on the course types and alternatives they stated to apply. The number of participants who received a question is stated in the results and indicated in brackets for figures and tables.

### 2.2. Survey Publication and Distribution

The anonymized, online German-language survey was published without attendance restriction from 31 May 2018 until 30 June 2019 [35]. In order to reach as many LAS trainers and supervisors as possible, the survey was presented at several international congresses including the European Society For Alternatives To Animal Testing (EUSAAT) Congress 2018 (Linz, Austria), 47th GV-SOLAS Seminar für Versuchstierkunde 2018 (Berlin, Germany), and the 3R-Tage Graz 2019 (Graz, Austria). Furthermore, it was published on online platforms such as the GV-SOLAS homepage [36] and distributed via personal e-mail invitations.

### 2.3. Data Analysis

During the survey period, Limesurvey software recorded 334 webpage views including 85 cases of survey participation, of which 38 were marked as “entirely completed”. Of these, 35 responses were used for analysis via IBM SPSS Statistics 25/26, as two were left entirely blank and one was handed in twice, one of which was omitted immediately after we received the notification. The 47 incomplete returns included 18 cases cancelled on page one, where the target group was selected by means of questions concerning their performed LAS courses. The second most frequent dropouts occurred on page four, concerning the awareness of simulators using a gallery including product names of all simulators and on page seven concerning practical training on live animals.

As the survey aimed at obtaining the first systematic insight into the current impact of the 3Rs on LAS education, a descriptive analysis was carried out individually for each variable. Questions aimed at the most frequent course type held by the individual respondent were first analyzed separately for each course type—FELASA B (now EU-Function A [6,29]), FELASA C (now EU-Function B [6,29]), and other course types. As no relevant differences were evident, the results of all course types were then analyzed collectively.

As this study focused on methodical and practical criteria for LAS courses, questions relating to anatomy were excluded from the analysis, which will need to be subjected to more detailed and specific analysis. The questions used for analysis are mentioned in the results in reference to the questionnaire (see Appendix A).

Data of open-ended questions were checked individually for each survey response as well as numerical input [37] for typos or non-causality. If values of numerical input were excluded due to typos or non-causality, it is noted accordingly in the results.

In the multiple choice and single selection questions, the absolute number of responses were determined. For numerical input, we calculated median value, minimum and maximum and for ranking questions, the mean value was defined to analyze the ratio between ranks.

Using a six-point Likert scale for assessment questions, the most commonly used approach in scaling survey answers [31,38], answers were assigned to two groups: rating values “1” to “3” represent positive assessment or agreement, whilst values “4” to “6” represent negative assessment or disagreement. The results are illustrated in Likert plots created by Microsoft Excel 2018.

## 3. Results

### 3.1. Demographics

The data revealed that 24 course trainers, eight supervisors and three other staff members took part in the survey. Of these, five stated that they were female, 24 male, two diverse, and four abstained. The median age of the respondents was given by 40.5 years and their experience in teaching LAS courses by 8.0 years. Appendix A provides an overview of the LAS courses regularly taught by the respondents (questions used for analysis see Appendix A question B1–B3, H3, U1–U4).

### 3.2. Training Courses on Rats and Mice

In order to obtain an overview of the most commonly practiced skills on animals, all respondents could indicate techniques, which were practiced on conscious, anaesthetized and/or dead animals in their most frequently performed course type (Table 2, questions used for analysis see Appendix A question H1, H2). The five most commonly practiced techniques on conscious rats and mice included handling, restraint, intraperitoneal, subcutaneous administration, and oral gavage. Except for the last two in conscious rats, all were chosen by more than two thirds of the respondents. In contrast, the three most commonly practiced techniques under anesthesia included cardiac blood sampling, subcutaneous administration, blood sampling via sublingual vein in rats, and cardiac blood sampling, cervical dislocation, and subcutaneous administration in mice. Training on cadavers of both species was stated by less than half of all respondents. The two most commonly practiced techniques on both species in this regard were suture techniques, for which many excellent alternatives are mentioned in several databases [39,40], and cervical dislocation.

### 3.3. Application of 3R Principle

Of all the possible alternatives included in the respondents’ (*n* = 35) most frequently performed course type, the implementation of simulators for rats and mice was relatively low, whereas soft toys and dolls were used twice as often (Figure 1, question used for analysis see Appendix A, question C1).

### 3.4. Awareness and Implication of Available Rat and Mouse Simulators

Under the assumption that using a simulator also entailed knowing a simulator, the results present a relatively high awareness of the simulators currently available. However, the analysis of the question “Which simulators are you familiar with?” revealed a very low implementation of them in the most frequently performed course type (Figure 2, questions used for analysis see Appendix A, question E1, E4). Rat simulator B was the only type used either sporadically or regularly in equal frequency of six respondents each—with a mean of two simulators per type. Beside rat simulator B, only rat simulator E was also used regularly by one respondent.

To indicate the average number of simulator types being used in LAS courses, the replies of the respondents who gave answers for all seven types currently available (*n* = 32 from total asked 35) were analyzed. The three respondents who gave answers only to a selection were omitted for this single analysis. On average, four types were known, but none was used. All types were known by seven respondents, one type—rat simulator B in all cases—by six respondents, and two respondents stated to know none. Four types of simulators were applied by one respondent, two types by two respondents and one type by ten respondents.

Fifteen of the total 35 respondents selected “sporadic or regular use” at least for one type of simulator in their most frequently performed LAS course. Nine respondents stated that they had used simulators predominantly for demonstration or practicing purposes parallel to the practice on live animals. A separate session for simulators was incorporated by six respondents and other modes of application were indicated by one respondent. Six of the 15 respondents stated that the exercise on simulators was voluntary, whilst five respondents indicated it being mandatory for all course participants. In three cases, simulator training was offered to individual participants (questions used for analysis see Appendix A question F1, F2).

Compared to the responses on rat simulators, which were chosen 8 times as an alternative method in LAS courses (Figure 1), two of them selected sporadic and five regular use of rat simulator B, while one of the latter additionally specified sporadic use of rat simulator A and one only stated regular use of rat simulator E (Figure 2). The remaining five statements on the use of rat simulator B (Figure 2) indicated that the simulator was used either only for testing purposes (*n* = 2) or only for demonstration or individual participants on rare occasions (*n* = 3). Due to lack of responses, further questions concerning certain types of simulators could be only analyzed for rat simulator B.

### 3.5. Satisfaction with the Course Application of Simulators

Of the 12 respondents who indicated the use of rat simulator B in their most frequently performed LAS course type, seven were generally dissatisfied whilst four were overall satisfied with its use. One did not give an answer (question used for analysis see Appendix A question I2). The analysis of the respondents’ evaluation of methodical characteristics indicated the highest rate of dissatisfaction with “oral gavage” and “restraint” whilst satisfaction was evident for “blood sampling” practice. Concerning handling and intravenous administration, the degree of satisfaction was very inhomogeneous amongst the respondents (Figure 3, questions used for analysis see Appendix A question I3, I4).

Beside methodical contentment, characteristics regarding the practical use were also assessed (Figure 4, question used for analysis see Appendix A question I6). On the one hand, the respondents were most unsatisfied with the material’s realism and the costs for acquisition and maintenance. On the other hand, easy handling including storage, transport, disinfection, and replacement of spare parts was overall approved of. Furthermore, the robustness of this type of simulator was also largely appreciated.

### 3.6. General Demand for a Novel Simulator

The general demand for a novel simulator among all respondents (*n* = 35) was analyzed separately for the twelve respondents planning or the 22 not planning on buying a new simulator. One respondent did not answer (Appendix A, question used for analysis see Appendix A question F3). The respondents in favor of buying a new simulator voted most for novel development. Interestingly, none stated to favor buying the same simulator he/she already possessed whilst three respondents were undecided (Appendix A, question used for analysis see Appendix A question F4). Furthermore, most of the respondents not planning on buying a new simulator indicated to be waiting for a new development. The second most frequent reasons included financial matters, inefficiency of simulators, and lack of application planned (Appendix A, question used for analysis see Appendix A question F5). Combining analysis from both groups, the majority of all respondents expressed a demand for new developments.

The transferability of techniques from rat simulator to mouse was analyzed independently (Appendix A). Twenty-two out of 35 respondents agreed that only certain techniques were transferable. One did not answer (Appendix A). More specifically, ear punch, subcutaneous, and/or intraperitoneal administration were considered the most transferable techniques but were not chosen by the majority of all respondents (Appendix A). As only a limited range of techniques may be considered transferable from rat simulator to mouse, the development of a separate mouse simulator must be considered (questions used for analysis see Appendix A question G1–G3).

### 3.7. Requirements for a Novel Simulator

All respondents (*n* = 35) were asked to determine methodical and practical requirements for a novel simulator (questions used for analysis see Appendix A question P1–P4, S1, T1). The five techniques ranked highest for a new rat simulator included handling, restraint, retro-bulbar blood sampling, intravenous administration, and oral gavage, whilst handling, restraint, oral gavage, intravenous and intraperitoneal administration ranked highest for a novel mouse simulator (Table 3, questions used for analysis see Appendix A question P1–P4). Analysis of practical requirements revealed that realistic material, followed by robust and disinfection-resistant material were the criteria most important to the respondents. Interestingly, none of the suggested technical prerequisites were irrelevant to the respondents overall (Appendix A, question used for analysis see Appendix A question S1). Furthermore, the respondents expressed their wish for a primarily realistic but also affordable, robust, and easy to handle simulator in an open field question concerning further methodical and practical requirements (Appendix A, question used for analysis see Appendix A question T1).

In feedback messages, respondents advocated simulator-based training for LAS courses and expressed their demand for a new rat and mouse simulator meeting their actual needs (Appendix A, question used for analysis see Appendix A question U5).

## 4. Discussion

For the protection of laboratory animals, the European Directive 2010/63/EU requires qualified education and training for respective persons prior to carrying out animal experiments [2].

Provided the required high standard of education is maintained or even improved by the use of simulators, they will simultaneously benefit the protection and well-being of the animals used for educational purposes as well as of those used in future experimental studies. However, studies specifically concerning simulator-based training of handling and procedural techniques on rats and mice in LAS education are lacking. Therefore, in a first step, this study aimed at analyzing the implementation and 3R impact of simulators on LAS training courses.

The results reflect the perspective of 35 trainers and supervisors of German-speaking countries. Several commonly practiced skills can be performed on the simulators currently available and the respondents seemed to be well aware of their availability.

However, they were marginally integrated in these courses. This could be due to the simulators not meeting the demands of the respondents. In total, the majority of the survey’s respondents expressed their wish for a novel, user-optimized, realistic, and financially affordable rat and mouse simulator.

### 4.1. Demographics

Courses accredited by FELASA [44] or GV-SOLAS [45] are listed online. Nevertheless, to our knowledge, information on the actual total number of LAS courses offered in Germany or Europe is lacking. Iatridou et al. performed a first data collection with regard to the number of European universities which include LAS education and training in under- or post-graduate curricula [46]. However, it did not include LAS education and training widely offered by scientific institutes, the pharmaceutical industry or service providers. Furthermore, statistical data on numbers of LAS courses and their respective trainers or supervisors, perhaps due to their diverse scientific backgrounds, seem to be lacking.

A total of 334 website views during the 13-month survey period reflects the high level of dissemination and interest in this study, even though we cannot exclude that one person visited the website or opened the survey repeatedly, which would be marked as incomplete return each time. Thus, to assure valid data, all 47 incomplete responses were excluded from analysis.

Participation was voluntary, not restricted, and anonymized. Despite this public conduction, we anyhow expect the results to be valid, as the questions and the respondents’ remarks contained LAS specific technical terms only understood with background expertise and each questionnaire response was checked individually. Furthermore, returns without answers or that were cancelled on page one—on which the most frequent LAS course type performed was to be selected by the participants in order to reach the target group—were excluded. Finally, 35 completed questionnaires were used for further analysis.

This sample size, in relation to the number of animals used per year [47,48,49,50], seems to be very low, although the actual total number of LAS course trainers and supervisors in German speaking countries is, to the authors’ knowledge, unknown to date.

We tried to reach most of the trainers and supervisors in German speaking countries by a public survey approach using several informative channels. Due this approach, we cannot exclude, that we, on the one hand, also reached scientists not involved in LAS education and training or interested laypeople or, on the other hand, missed trainers and supervisors who did not participate on the congresses nor read the information on the website used for advertisement. Furthermore, we cannot exclude that we reached only a certain group of LAS trainers and supervisors being particularly interested in simulators. Therefore, the results reflect a certain point of view on the current situation of simulators.

As courses in German speaking countries are often structured similarly or set up according to the recommendations by the GV-SOLAS, the questionnaire was exclusively in German aiming at German speaking countries to provide the necessary basis of comparability which was essential to our study. As we probably missed non-German speaking course trainers and supervisors in German speaking countries, an additional English questionnaire might have reached more participants, providing more data. Moreover, future studies in other languages may aid in collecting additional data from other countries.

To increase the target population, cooperation with LAS education and trainings committees, e.g., FELASA [51,52], International Council for Laboratory Animal Science (ICLAS) [53] or Education and Training Platform for Laboratory Animal Science (ETPLAS) [54], might have led to more comprehensive data and study expressiveness, which could be considered for following projects. Furthermore, a non-public survey approach using mandatory questions instead of a combination with voluntary and conditional questions may lead to more valid data. Nevertheless, this study gives an initial overview of LAS course trainers’ and supervisors’ current awareness, satisfaction, and implementation regarding simulators and provides incentive and indications for further studies to promote humane teaching and training.

### 4.2. Training Courses on Rats and Mice

The simulators currently available covered most of the techniques stated to be practiced on live animals by the respondents, e.g., handling, restraint, and oral gavage. In this regard, the simulators may therefore have a great potential affecting the 3Rs in LAS education.

As the available simulators each cover a different range of techniques, several simulator types would need to be purchased in order to cover all possible training techniques. Thus, each course participant would ideally need to practice on all available simulators prior to exercise on live animals in order to provide an identical training for everyone. Acquisition costs for one simulator lie in the upper three- to lower four-digit Euro range and may therefore increase the financial burden inflicted by simulator-based training, depending also on the number of simulators acquired for the courses. Financial matters may be one of the factors affecting the limited implementation of simulators in LAS courses, as mentioned in prior studies on other alternatives used in teaching [15,18,55]. The impact of financial issues was also reflected by the respondents who overall chose the lack of financial resources as the second most important reason for not being in favor of buying a simulator (see Appendix A).

Besides financial considerations, lack of objective data, pre-existing knowledge, experience of colleagues and students, and personal attitudes were also identified as main factors influencing the introduction of alternative learning methods in biology and scientific education [15,55,56]. These factors were therefore also considered in this survey study.

### 4.3. Application of 3R Principle

In comparison to other non-animal alternatives, the application of simulators appeared to be poor in this survey.

However, it must be considered that each device has specific learning outcomes leading to a rather complementary than interchangeable application of alternatives. Instructions, photos, drawings, and video material [57,58] are mainly used as supportive or preparatory material to be combined with other alternatives, as they provide a good theoretical background by passive education [23] (pp. 9–10). Learning software [59,60] with passive and active elements may also be regarded as a good alternative in theoretical preparation for exercises [61,62,63]. Next, computer simulations, classified as passive or active educational measures may be considered beneficial [10,56]. However, they are not capable of imitating physical reactions of the live animal and therefore do not cover all skills essential for animals experiments [56]. Furthermore, they are mostly adapted to the designer’s special needs and software [10,56] and, to our knowledge, there is no simulation available for the purpose of practicing handling and procedures in rats and mice. Virtual reality (VR) simulators, especially those including haptic feedback, bear great potential in LAS practical training as haptic feedback is essential for many LAS-related procedures. Although some medical simulators for surgical techniques [64,65] and veterinary simulators for several animal species are available [66,67,68,69,70], haptic VR simulators appropriate for practicing handling and minimally invasive techniques on small mammals, e.g., rats and mice, do not exist to the authors knowledge to date.

In summary, these alternative methods have limitations in providing comprehensive transferable skill acquisition of animal handling and procedures. This may be accomplished by toys, suture pads [71], silicon ears [72], and simulators [23] (pp. 9–10). Specialized material such as suture pads or silicone ears are limited to training of certain techniques, whereas simulators can provide a range of different procedural techniques (see Table 1) and may therefore provide the most comprehensive skill acquisition prior to practice on live animals.

According to previous studies concerning alternatives [15,18,55,56], the use of suture pads may be higher than for simulators, because nowadays a lot of experience and scientific data on them exist [73,74,75], as they have been used as a non-animal alternative for many years. Additionally, the consumer market as well as the market for specialized medical training aids provide a range of suture pads for diverse applications in different price ranges so that information on them is very easily accessible.

### 4.4. Awareness and Implication of Available Rat and Mouse Simulators

The survey’s analysis revealed a high awareness of existing simulators amongst the respondents and, as expected, the simulator that has been available longest, rat simulator B, was known to most of the respondents. Taking into account that the results may only reflect a certain point of view due to the small sample size used for this study, the degree of awareness determined in this study seems plausible as the simulators are mentioned in publications on teaching alternatives [23,24,25] and reports of new technologies [26] or introduced in databases for alternative learning, e.g., InterNICHE—International Network for Humane Education [76] or NORINA—A Norwegian Inventory of Alternatives [27] which can be assumed to be largely known by LAS scientists. Additionally, the trainers’ and supervisors’ interest in simulators and simulator-based training may have increased due to the previous project advertisement.

On the opposite, based on the study results, a marginal role of all currently available simulators in alternatives for LAS courses may be assumed. This was firstly indicated by the small response rate for simulator implementation in courses. Secondly, this was also indicated by these respondents indicating a predominant use for demonstrations and voluntary training in parallel to the exercise on animals. In our study, further analysis of the absolute number of simulators used per course or concerning satisfaction was possible only for rat simulator B, as this simulator was the only type used regularly or sporadically by one third of all respondents. A simulator-to-participant ratio of a mean of two rat simulators per type B in courses with a median of seven to 20 people seems very low (Appendix A).

In further studies, analysis of existing knowledge and implementation via two separate questions may increase data validity. Furthermore, a third response option “temporary usage” could be added to the question item regarding the application of simulators, in order to enable a comparison between the perspectives of respondents who had used simulators in the past versus those currently using them.

### 4.5. Satisfaction and Requirements Regarding a Novel Simulator

In general, dissatisfaction with the simulators available was evident. This was indicated by the low rate of simulator implementation and the respondents’ criticism concerning the current situation based on several open text remarks or answer selections. As only respondents stating to currently use simulators were able to assess these, the general satisfaction may be lower than presented in this study.

Ideally, all techniques should be practiced on simulators prior to carrying out animal-based training. However, the higher the number of techniques implemented in the design of a novel simulator, the more complex and higher-priced the simulator will be, which, in turn, might impact the usage [15,18,55].

As expected, the 10 techniques ranked highest included techniques most commonly practiced on live and particularly on conscious animals, e.g., handling or restraint, techniques for which no simulator is currently available, e.g., blood sampling via retro-bulbar plexus for rats or subcutaneous administration for mice and techniques assessed not satisfactory by the survey’s respondents, e.g., oral gavage for rat simulator B. Unexpectedly, cervical dislocation, a common technique for humanely sacrificing mice, was not amongst the top ten, although it was the second most commonly practiced technique on anesthetized and dead mice in the most frequently employed course type, but microsurgical techniques were, which can be considered very specific and thus rarely performed (see Table 2). Interestingly, intravenous administration via tail vein, which is not one of the most commonly practiced techniques, but which is covered by nearly all simulators, was also amongst the top five. Therefore, these techniques should be favored for a novel simulator.

As rat simulator B provides some techniques that are in high demand for a novel simulator, e.g., restraint and oral gavage, the simultaneous overall negative methodical assessment of these techniques revealed that the simulator itself did not fulfil the needs of the respondents.

In contrast, the assessment of requirement versus satisfaction regarding handling and intravenous administration seemed to differ greatly amongst the respondents. Concerning handling, for example, some respondents were very dissatisfied, whereas a close majority assessed it positively.

Next, all requirements were determined of overall importance, with high emphasis on realistic material, usability, and cost-effectiveness. The significant role of realism for validity was also reviewed by Zemanova and Knight [18]. In comparison to the practical characteristics, rat simulator B did not seem to meet the needs of realistic material and cost-effectiveness, but of usability and durability.

### 4.6. General Demand for a Novel Simulator

In total, the majority of the survey’s respondents expressed their wish for a novel, user-optimized, realistic, financially affordable, and robust rat and mouse simulator. Concerning the simulator species, a demand for both a novel rat and mouse simulator was evident.

Although almost two thirds of the respondents stated that certain techniques applied to mice could be trained on a rat simulator, there was no absolute consensus on the individual transferable techniques. The use of a rat simulator for training mouse-related techniques as possible measure for cost reduction and time efficiency bears several important drawbacks. First, because the participants are inexperienced, a transfer between species may lead to handling difficulties due to differences in size and techniques of restraint. Furthermore, confusion may arise concerning the type of techniques carried out exclusively on one particular species. Hence, training techniques for a certain species using non-species-specific simulators may not be favorable for all techniques in terms of learning outcome. If the use of non-species-specific simulators is considered, techniques and learning outcome must be critically determined prior to their implementation and carefully evaluated during use.

## 5. Conclusions

The respondents’ positive attitude towards simulator-based training in LAS courses and their demand for novel and particularly realistic simulators displayed the necessity of further evaluations.

Therefore, a detailed anatomical analysis shall be carried out by laboratory animal scientists and veterinary anatomists by which anatomical requirements will be determined scientifically. The second most important determinant affecting the implementation of simulator-based training are the LAS course participants themselves. They are the only ones capable of assessing the simulators’ practical learning efficiency. As this defines the impact on the 3Rs, all simulators currently available may be evaluated in a broad German and English survey by LAS course participants, also particularly concerning requirements for simulator-based training.

The results from these following studies shall then be integrated into a comprehensive specification analysis for a novel simulator optimally adapted to the diverse requirements in LAS courses. This multi-dimensional analysis may set the ground for a simulator-based training, which may guarantee high-quality education and thereby services the well-being and protection of all laboratory rats and mice—in education and in future experiments.

## Figures and Tables

**Figure 1 animals-11-01848-f001:**
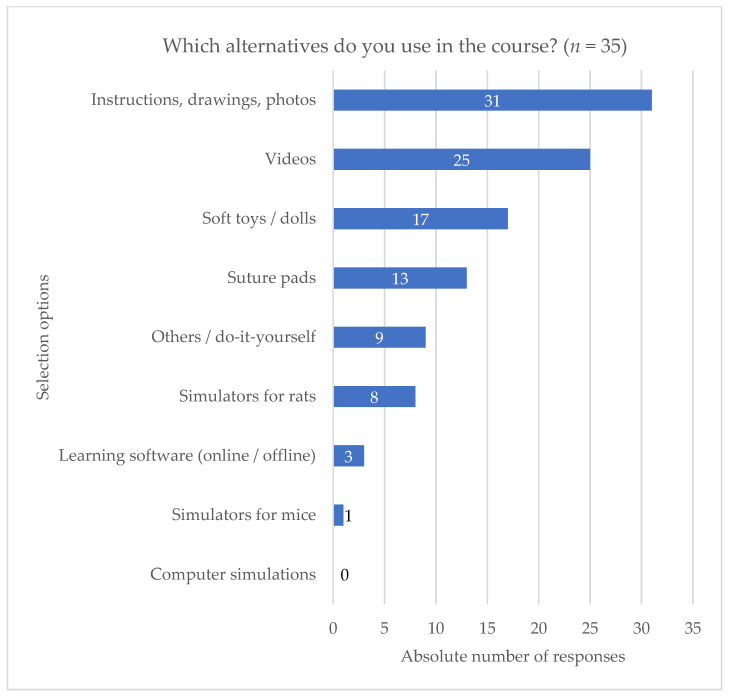
Descriptive analysis of the use of alternatives in training courses. Distribution of replies concerning the type of alternatives used in the most frequently performed course type derived from the multiple-choice question “Which alternatives do you use in the course?”. Absolute number of responses is shown for each respective method (*n* = 35).

**Figure 2 animals-11-01848-f002:**
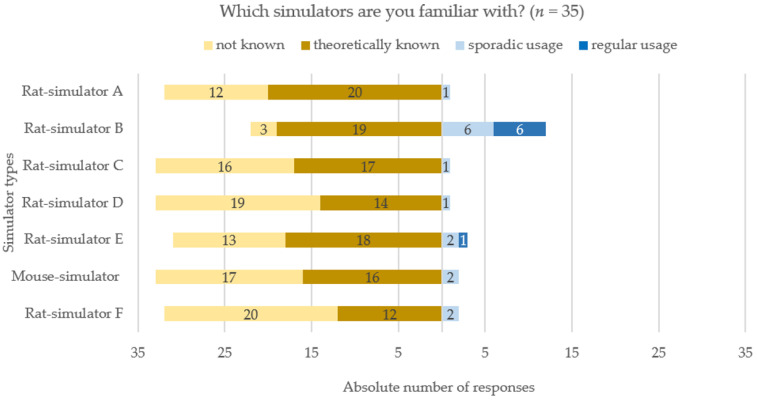
Descriptive analysis of the awareness and usage of currently available rat and mouse simulators derived from the single selection question “Which simulators are you familiar with?” for each simulator type. Distribution of replies concerning the type of rat or mouse simulator the respondents were familiar with, referring to the most frequently performed course type. Data are shown in a diverging stacked bar chart. Simulators are listed on the y-axis, response frequencies on the x-axis, ranging from “not known” (yellow) and “theoretically known” (brown) on left- to “sporadic usage” (light blue) and “regular usage” (dark blue) on the right-handed diverging stacked bars. Single selection question for each simulator type. Absolute number of responses is shown for each respective simulator in the corresponding bar (*n* = 35).

**Figure 3 animals-11-01848-f003:**
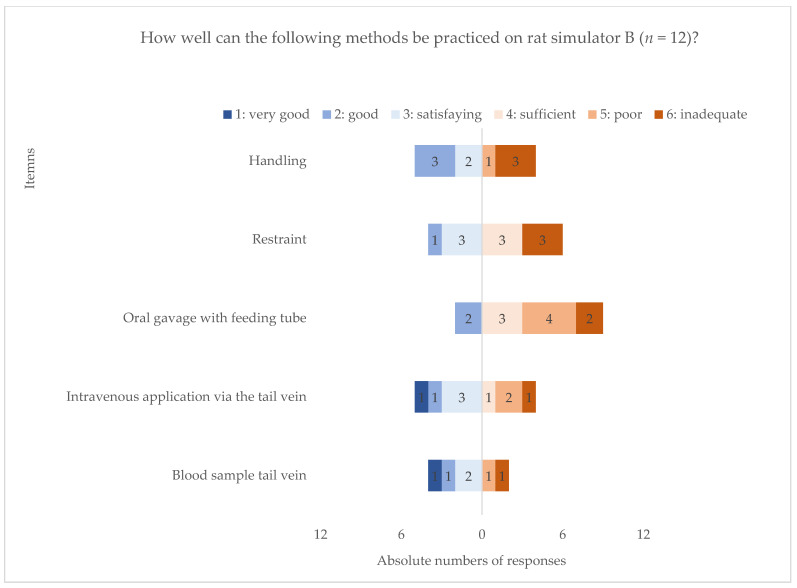
Descriptive analysis of methodical satisfaction with rat simulator type B. Distribution of replies on how well certain handling techniques can be practiced on rat simulator type B on a six-point Likert scale, referring to the most frequently performed course type. Data are shown in a diverging stacked bar chart. Handling and procedural techniques are listed on the y-axis, response frequencies on the x-axis, ranging from “1: very good” (dark blue), “2: good” (blue), “3: satisfactory” (light blue) on left- to “4: sufficient” (light red), “5: poor” (red), “6: inadequate” (dark red) on the right-handed diverging stacked bars. Absolute number of responses is shown for each technique in the corresponding bar (*n* = 12).

**Figure 4 animals-11-01848-f004:**
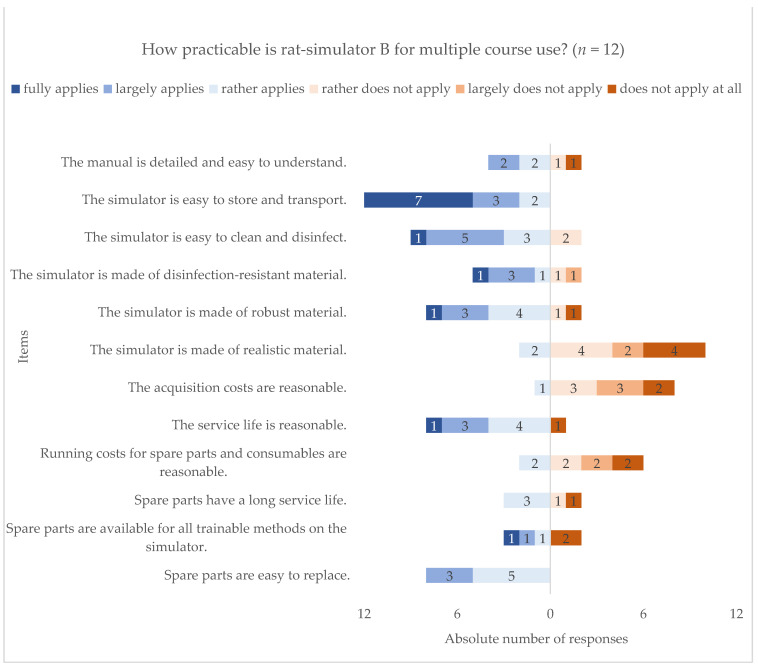
Descriptive analysis of the practical satisfaction with rat simulator type B. Distribution of replies on the practicability of rat simulator B for multiple use on a six-point Likert scale, referring to the most frequently performed course type. Data are shown in a diverging stacked bar chart. Statements about practicability are listed on the y-axis, response frequencies on the x-axis, ranging from “1: fully applies” (dark blue), “2: largely applies” (blue), “3: rather applies” (light blue) on left- to “4: rather does not apply” (light red), “5: largely does not apply” (red), “6: does not apply at all” (dark red) on the right-handed diverging stacked bars. Absolute number of responses is shown for each statement in the corresponding bar (*n* = 12).

**Table 1 animals-11-01848-t001:** Overview of available rat and mouse simulators for handling and procedural techniques, and for microsurgical techniques (2018). Product names were anonymized.

**Product Information**	**Available Simulators for Handling and Procedural Techniques (2018)**					
	**Rat Simulator A**	**Rat Simulator B**	**Rat Simulator C**	**Rat Simulator D**	**Rat Simulator E**	**Mouse Simulator**
External appearance	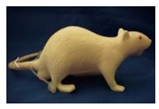	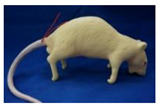	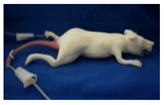	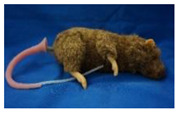	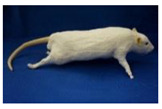	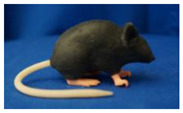
Norina database record number [27]	5e236	88cf4	05ebd	457b1	f7a0d	45635
Techniques to be trained ^1^	Handling Restraint	Handling Restraint	Handling Restraint	Handling Restraint	Handling Restraint	Handling Restraint
	Administration -by oral gavage -intravenous via tail vein	Administration -by oral gavage -intravenous via tail vein	Administration -intravenous via tail vein	Administration -intravenous via tail vein	Administration -by oral gavage -intravenous via tail vein-subcutaneous (neck/flank) -intramuscular	Administration -by oral gavage ^2^ -intravenous via tail vein -intraperitoneal ^3^
	Blood sampling via -tail vein	Blood sampling via -tail vein	Blood sampling via - tail vein -saphenous vein -cardiac blood sampling	Blood sampling via -tail vein	Blood sampling via -tail vein	-
	Endotracheal intubation	Endotracheal intubation	Endotracheal intubation	Ear punch	Micro-chipping Temperature measurement	-
Specifications	Performance control fororal gavage	Performance control fororal gavage	-	Moveable head Material with fur	Flexible joints: jaw, spine, and legs	Claws on front paws
Spare parts	Tail Artificial blood	Tail Artificial blood	Artificial blood	Tail Artificial Ears Artificial blood	Tail Artificial blood Blood sampling kit	Tail Powder Lubricant
**Product Information**	**Available Simulator for Microsurgical Techniques (2018)**					
	**Rat Simulator F**					
External appearance	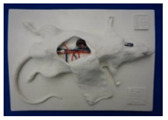					
Norina database record number [27]	4bcdc					
Techniques to be trained ^1^	Anastomosis of vessels and organs					
	Cannulations					
	Transplantations					
Specifications	-					
Spare parts	Set of vessels and organs					

^1^ Techniques for training according to the manual. ^2^ Oral gavage carried out with lubricant bottle. ^3^ Intraperitoneal administration without needle injection (only with syringe).

**Table 2 animals-11-01848-t002:** Descriptive analysis of techniques trained on live animals in Laboratory Animal Science Courses. Distribution of replies concerning techniques practiced on conscious or anesthetized rats and/or mice or post mortem in the most frequently performed course type derived from the multiple choice question “Which techniques do you train on animals in the practical course part? Please indicate whether training is done on conscious or anesthetized rat or mouse or post mortem”. Absolute number of responses are shown for each technique and species with regard to its condition (*n* = 35).

Procedural Category	Technique	Absolute Number of Responses (*n* = 35)					
		Practice on Rats			Practice on Mice		
		Conscious	Anesthetized	Post Mortem	Conscious	Anesthetized	Post Mortem
Husbandry and breeding techniques	Handling	29	9	3	34	11	5
	Restraint	29	10	4	34	11	5
	Ear punch	7	6	5	11	8	7
	Vaginal cycle control	2	2	0	2	2	0
Administration techniques	Oral gavage	17	6	2	31	8	1
	Oral, voluntary	9	1	0	9	1	0
	Subcutaneous	18	15	3	26	18	3
	Intramuscular ^1^	4	11	3	5	9	2
	Intraperitoneal	23	10	2	33	10	2
	Intravenous via dorsal penis vein ^1^	0	4	0	0	2	0
	Intravenous via tail vein	5	10	1	12	12	1
Blood sampling via	Sublingual vein ^2^	2	12	1	0	4	0
	Facial vein ^3^	0	2	0	19	14	2
	Retro-bulbar plexus	0	10	0	1	17	4
	Saphenous vein	0	10	1	10	9	0
	Tail vein	4	11	0	8	13	0
	Cardiac blood sampling ^4^	0	16	5	0	23	9
Surgical techniques	Endotracheal intubation ^5^	0	2	1	0	1	1
	Microsurgical techniques ^5^	0	8	5	0	8	4
	Suture techniques ^5^	0	7	9	0	11	12
Euthanasia	Cervical dislocation ^6^	0	4	7	13	22	10
Miscellaneous	Other techniques	2	5	4	3	6	4

^1^ Not recommended for mice [41]; ^2^ not recommended for mice [42]; ^3^ not recommended for rats [42]; ^4^ not on conscious animals [42]; ^5^ not on conscious animals [43]; ^6^ only ≤1000 g body weight; sedation required for animals >150 g body weight [43].

**Table 3 animals-11-01848-t003:** Descriptive analysis of methodical requirements for a novel simulator. Distribution of replies (*n* = 35) concerning the top 10 techniques most desired to be practiced on a novel rat or mouse simulator as ranking question (highest priority at top). A maximum of 10 out of 20 suggested answers could be ranked. Data are shown as calculated mean values for each technique and species.

Methodical Requirements for a Novel Simulator (*n* = 35)				
Rank	Techniques for a Novel Rat Simulator	Calculated Mean Values (Standard Deviation σ) of Techniques in Rats	Techniques for a Novel Mouse Simulator	Calculated Mean Values (Standard Deviation σ) of Techniques in Mice
1	Handling	2.94 (±3.03)	Handling	1.92 (±1.65)
2	Restraint	3.78 (±3.30)	Restraint	3.24 (±2.91)
3	Blood sampling via retro-bulbar plexus	3.80 (±2.31)	Oral gavage	3.93 (±2.07)
4	Intravenous administration via tail vein	4.05 (±2.01)	Intravenous administration via tail vein	4.56 (±2.09)
5	Oral gavage	4.31 (±2.23)	Intraperitoneal administration	4.92 (±2.08)
6	Blood sampling via tail vein	4.80 (±2.73)	Subcutaneous administration	5.13 (±2.36)
7	Oral administration, voluntary	5.00 (±1.63)	Microsurgical techniques	5.17 (±3.62)
8	Intraperitoneal administration	5.30 (±1.45)	Blood sampling via retro-bulbar plexus	5.29 (±2.22)
9	Microsurgical techniques	5.34 (±5.33)	Blood sampling via facial vein	5.35 (±1.90)
10	Blood sampling via saphenous vein	5.44 (±3.40)	Blood sampling via tail vein	5.39 (±2.43)

## Data Availability

The data presented in this study are available on request from the corresponding author. The data are not publicly available due to unanalyzed data not related to the subject of this study.

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
