# Peer review of "Alternatives in Education—Rat and Mouse Simulators Evaluated from Course Trainers’ and Supervisors’ Perspective"

_animals, 2021, doi:10.3390/ani11071848_

Round 1

Reviewer 1 Report

Humpenöder and colleagues deals with an interesting argument: the trainers’ point of view regarding the simulator-based training in LAS courses. The manuscript is well written and the authors use a qualitative approach to investigate the aforesaid topic. However, there are few points that need to be clarified, especially with regards to material and methods:

  1. P1, L 37-41: Looking at the abstract only, I was initially a bit confused whether the survey of the current study was provided to several categories of participants (e.g. course trainers, course participants, experts etc.) or to course trainers only. I had to read both Introduction and M&M sections to understand that this manuscript presented the results of part of a larger study (i.e. the SimulRATor project) and that the participants of the current study were the course trainers only. Am I correct? If so, I would suggest to focus only on the aim of the current study - both in the abstract and in the Introduction sections - to avoid confusion, while eventually providing few more info about the whole project, if needed, in the discussion section.
  2. P1, L 41: Replacing ‘these’ with ‘simulators’ may help to improve the flow of the sentence.
  3. P5, L 124: As human participants are involved in the study, did the authors obtain an ethical approval by the Ethics Committee prior the starting of the study?
  4. P2, L 79-84: Please consider to re-phrase this sentence. Too long.
  5. P3, L 100-113: This is what I meant in my previous comment. At the end of a well-written Introduction, the readers suddenly find ‘Hence, this (Did you talk about it before?) SimulRATor project..’ and the list of the 3 aims of the project. In my opinion, being focused on the aim of the current study can help to avoid confusion.
  6. P5, L 138: (Table S1) - I agree with the authors about the importance of enclosing the questionnaire as supplementary material. However, given the international audience of the Journal, I would suggest to provide an English version of the questionnaire.
  7. P5, L 139-140: Do the authors mean that the participants did not receive/answer to the same number and type of questions? If so, why did you decide to apply this approach? I understood that you carried out a survey with closed-ended questions not a semi-structured interview approach. Please, clarify this aspect.
  8. P5, L 149: ‘LAS trainers and supervisors as possible,..’. Sometimes the authors refer to ‘trainers and supervisors’ while sometimes just ‘trainers’. Please, be consistent throughout the text.
  9. P6, L 175-182: This comment is linked to my previous one (N. 7). Here again I wonder whether you used a mixed of closed and open-ended questions? If so, please provide more details on the classification/number of questions and type of approach used. Also, Table 3 at page 12 (L 330) of the manuscript presents the rank of techniques according to the ‘calculated mean values’. Did the authors calculate these values based on the 6-point Likert scale assigned a posteriori and described in L 175-179? Why did you decide to make this calculation only for the data presented in Table 3 and not for all the other questions where the Likert scale was used?
  10. P8, L 243: Do the authors mean ’32 respondents’ and not ‘35’? Few lines above (L 237-238) they said that 3 were omitted from the analyses.
  11. P13, L 350-351: Please consider to re-phrase this sentence.
  12. P12, L 370-371: The authors here say that they ‘addressed most of the German and many European professionals’ while few lines later (L 383-384) they state that ‘the questionnaire was exclusively in German aiming to German speaking countries’. This seems a contradiction to me, did you target German only professionals or European professionals too? Please, clarify this aspect.
  13. P 12, L 378-379. This seems a possible bias towards trainers more interested in simulators. Thus, how can you be sure that you ‘addressed most of the professionals with educational responsabilities’ as stated in L 368-371? Consider to discuss possible disadvantages of the study.
  14. P14, L 395: Typo. Remove the final ‘s’ from ‘to exercises’.

Reviewer 2 Report

Congratulations authors, you have undertaken important research looking at the 3R's using rat and mouse simulators.  I enjoyed reading your paper.

I would recommend revising Figure 2 as the order of simulator types on the Y axis wasn't clear to me.  It wasn't clear to me why you didn't follow the order used in the photographs, namely, Rat A, B, C etc.

I also recommend having a native English speaker read over your draft as they may provide some helpful comments to improve readability. 

Good luck with publication and with your important work in alternatives to the harmful use of animals in research. 

Reviewer 3 Report

This paper describes a small-scale survey (n=35) of trainers in LAS courses regarding the use of models/simulators as teaching aids. Understanding the needs of educators in this regard are likely to result in reduction in animal numbers and refinement of the procedures, and therefore promote the welfare of the animals.

In general, I found the Introduction and Methods to be very well written. The Results, and to a larger extent, the Discussion, however, require some modifications. It feels as if two different people wrote the different sections.

My detailed comments are as follows:

Introduction:

56-57: can you please demonstrate how high educational standards are likely to improve reproducibility? I understand your intent here, but I think that the connection needs to be specified.

79-83: please provide references or indicate that these statements are based on personal experience.

111-112: please provide a reference for specification analysis.

Methods:

2.3 Data analysis: it would be helpful to know how big the sample size in comparison to the entire population is.

Results:

3.1 Demographics: please add educational background and professional experience.

3.3 Use of 3R methods:  I think that what you are alluding to here is the incorporation of the principle of replacement. If that’s the case, why not simply state that, instead of the vaguer terminology you are currently using (3R methods)?  In fact, the 3Rs are principles rather than methods.

238-241: this paragraph can be improved. Please rephrase.

244-250: Same with this paragraph. It is very cumbersome.

Table 3.: please add the standard deviations.

Figure 3: please change handing to handling.

Discussion:

350-351: This sentence reads a bit funny. Please rephrase.

396: Can you please provide a range of the cost of the different simulators to give the reader a clearer view of the financial burden?

4.3: please see my comment for 3.3.

406-409: this paragraph is unclear.

437: please clarify what you mean by the specialized trade.

443-444: are you referring to all other simulators (apart from B)? It is not clear.

445: what previous survey? Is there a reference for that?

441-456: this entire paragraph needs work. I was unable to follow any logic here.

467-469: please rephrase.

Round 2

Reviewer 1 Report

I would like to thank the authors for their work. They addressed all my comments and improved the paper, thus I consider their manuscript suitable for publication.

I have just few typos to highlight:

  1. P13, L578: 'the numbers of of simulators' should be 'the number of simulators';
  2. P13. L581-582: Do the authors mean '..who overall chose the lack of financial resources as the second most important reason for not being..'?
  3. P15, L 739: 'comparison' instead of 'comparability' ?

Author Response

Please find the reply in attachment
